# Comparison between [^68^Ga]Ga-PSMA-617 and [^18^F]FET PET as Imaging Biomarkers in Adult Recurrent Glioblastoma

**DOI:** 10.3390/ijms242216208

**Published:** 2023-11-11

**Authors:** Caterina Brighi, Simon Puttick, Amanda Woods, Paul Keall, Paul A. Tooney, David E. J. Waddington, Vicki Sproule, Stephen Rose, Michael Fay

**Affiliations:** 1Image X Institute, Faculty of Medicine and Health, Sydney School of Health Sciences, The University of Sydney, Sydney 2015, Australia; paul.keall@sydney.edu.au (P.K.); david.waddington@sydney.edu.au (D.E.J.W.); 2AdvanCell Isotopes Pty Ltd., Sydney 2000, Australia; simon@advancell.com.au (S.P.); stephen@advancell.com.au (S.R.); 3GenesisCare, Newcastle 2290, Australia; amanda.woods@genesiscare.com (A.W.); vicki.sproule@genesiscare.com (V.S.); mikefay@me.com (M.F.); 4MHF Centre for Brain Cancer Research, College of Health, Medicine and Wellbeing, University of Newcastle, Newcastle 2308, Australia; paul.tooney@newcastle.edu.au

**Keywords:** [^68^Ga]Ga-PSMA-617, recurrent glioblastoma, positron emission tomography imaging of glioblastoma, theranostics for glioblastoma

## Abstract

The aim of this prospective clinical study was to evaluate the potential of the prostate specific membrane antigen (PSMA) targeting ligand, [^68^Ga]-PSMA–Glu–NH–CO–NH–Lys-2-naphthyl-L-Ala-cyclohexane-DOTA ([^68^Ga]Ga-PSMA-617) as a positron emission tomography (PET) imaging biomarker in recurrent glioblastoma patients. Patients underwent [^68^Ga]Ga-PSMA-617 and O-(2-[^18^F]-fluoroethyl)-L-tyrosine ([^18^F]FET) PET scans on two separate days. [^68^Ga]Ga-PSMA-617 tumour selectivity was assessed by comparing tumour volume delineation and by assessing the intra-patient correlation between tumour uptake on [^68^Ga]Ga-PSMA-617 and [^18^F]FET PET images. [^68^Ga]Ga-PSMA-617 tumour specificity was evaluated by comparing its tumour-to-brain ratio (TBR) with [^18^F]FET TBR and its tumour volume with the magnetic resonance imaging (MRI) contrast-enhancing (CE) tumour volume. Ten patients were recruited in this study. [^68^Ga]Ga-PSMA-617-avid tumour volume was larger than the [^18^F]FET tumour volume (*p* = 0.063). There was a positive intra-patient correlation (median Pearson r = 0.51; *p* < 0.0001) between [^68^Ga]Ga-PSMA-617 and [^18^F]FET in the tumour volume. [^68^Ga]Ga-PSMA-617 had significantly higher TBR (*p = 0*.002) than [^18^F]FET. The [^68^Ga]Ga-PSMA-617-avid tumour volume was larger than the CE tumour volume (*p* = 0.0039). Overall, accumulation of [^68^Ga]-Ga-PSMA-617 beyond [^18^F]FET-avid tumour regions suggests the presence of neoangiogenesis in tumour regions that are not overly metabolically active yet. Higher tumour specificity suggests that [^68^Ga]-Ga-PSMA-617 could be a better imaging biomarker for recurrent tumour delineation and secondary treatment planning than [^18^F]FET and CE MRI.

## 1. Introduction

Prostate-specific membrane antigen (PSMA) is a cell-surface protein that is highly expressed in prostate cancer [1] and also in the endothelium of the tumour neovasculature of several solid tumours [2], including glioblastoma (GBM) [3,4,5,6], making it an excellent target for antibody–drug conjugates or peptide receptor radionuclide therapy [3].

Recently [^68^Ga]-PSMA–Glu–NH–CO–NH–Lys(Ahx)-HBED-CC ([^68^Ga]Ga-PSMA-11) has also been shown to be a promising positron emission tomography (PET) imaging agent for the diagnosis of patients with primary GBM [7] and for treatment response assessment [8,9] and, as such, it has great potential as an imaging biomarker for management of GBM patients [7]. Modifications of the PSMA-11 peptide have resulted in the development of the small-molecule ligand PSMA-617, which has the advantage of being suitable for radiolabelling with different radioisotopes, including gallium-68, lutetium-177, indium-111 and yttrium-90, and thus it can be used for both PET imaging (gallium-68, yttrium-90) and radionuclide targeted therapy (lutetium-177, indium-111, yttrium-90) [10]. [^68^Ga]-PSMA–Glu–NH–CO–NH–Lys-2-naphthyl-L-Ala-cyclohexane-DOTA ([^68^Ga]Ga-PSMA-617) has a lower non-specific affinity to the kidneys compared to [^68^Ga]Ga-PSMA-11 [10], and preliminary clinical results show significantly high values of in vivo uptake into high grade gliomas [11]. These characteristics make PSMA-617 one of the best PSMA-targeting ligand candidates for theranostics applications [10].

In a recent clinical study Kunikowska et al. [12] postulated a potential unsuitability of PSMA-11-based targeted therapies in recurrent GBM patients, due to the low median [^68^Ga]Ga-PSMA-11 tumour-to-liver ratio (TLR) observed in their patient cohort (i.e., 0.8, range: 0.6–1.8, *N* = 15). However, the same authors published promising results of a case study of a recurrent GBM patient who had a [^68^Ga]Ga-PSMA-11 maximum standardised uptake value (SUV_max_) = 10.3 in the tumour lesion with a TLR of 1.8 and underwent [^177^Lu]Lu-PSMA-617 treatment as part of a dosimetry study [13]. The study demonstrated that the absorbed dose in the tumour was 100 times higher than in the kidneys, 8.4 times higher than in the liver and 28.7 times higher than in the whole body, and that the radiotracer had a very slow washout in the tumour compared to the other organs in the days following administration [13], generating interest in testing the feasibility of using PSMA-617-based targeted therapy with α/β-emitters for the treatment of recurrent GBM patients.

Our pilot clinical trial aims to evaluate the potential of [^68^Ga]Ga-PSMA-617 as a PET imaging biomarker and, as an exploratory aim, evaluate its potential as a candidate ligand for targeted radionuclide therapy in patients with recurrent GBM. In this study, PET-CT imaging was used to compare the tumour uptake of [^68^Ga]Ga-PSMA-617 with the uptake of O-(2-[^18^F]-fluoroethyl)-L-tyrosine ([^18^F]FET), an amino acid that has much higher tumour specificity than [¹⁸F]Fluorodeoxyglucose ([^18^F]FDG) in brain cancers [14,15]. The superior ability of [^18^F]FET PET to estimate gross tumour volume and to image areas of metabolically active residual infiltrating tumour post-surgery is due to the tumour-targeting mechanism of FET, which relies on the transport across vascular membranes by system L-type amino acid transporters and is, therefore, not affected by the presence of an intact blood–brain barrier [16]. As a consequence, [^18^F]FET PET is used in this study to define the extent of gross tumour mass and levels of tumour metabolic activity. By comparing measurements of biological tumour volume (BTV) and tumour-to-brain ratio (TBR) obtained with [^68^Ga]Ga-PSMA-617 and [^18^F]FET PET images, this study aims to determine [^68^Ga]Ga-PSMA-617 tumour selectivity and specificity, respectively. Moreover, by measuring standardised uptake values (SUV) of [^68^Ga]Ga-PSMA-617 in the tumour and in the healthy contralateral brain tissue, TLR and tumour-to-salivary glands (TSG) ratio, this study aims to explore the potential of PSMA-617 as a candidate ligand for targeted radionuclide therapy with α/β-emitters in patients with recurrent GBM.

## 2. Results

### 2.1. Patient Recruitment and Data Collection

Ten recurrent GBM patients were recruited between October 2018 and September 2021. Pre-enrollment T_1_-weighted contrast-enhanced (T1CE) MR images were provided by the original site of recurrence diagnosis for nine patients, [^18^F]FET and [^68^Ga]Ga-PSMA-617 brain images were collected for ten patients and [^18^F]FET PET scan details needed for the conversion of [^18^F]FET PET images into SUV were collected for nine patients. [^68^Ga]Ga-PSMA-617 PET images were acquired as full-body scans for only two patients. Dates of the PET scans for each patient are reported in Table 1. The median interval between [^18^F]FET and [^68^Ga]Ga-PSMA-617 PET scans was 5 days (range: 3–15). PET imaging metrics extracted from the data are collected in Table 2, Table 3 and Table 4.

### 2.2. Evaluation of [^68^Ga]Ga-PSMA-617 Tumour Selectivity

The results of the comparison of the tumour volume between the [^68^Ga]Ga-PSMA-617 and [^18^F]FET BTV are shown in Figure 1. The mean BTV delineated on [^68^Ga]Ga-PSMA-617 PET images was slightly larger, although not significantly (*p* = 0.063), than the respective mean BTV delineated on [^18^F]FET PET images. This result was further confirmed by the measurement of the mean volumetric ratio between [^68^Ga]Ga-PSMA-617 BTV and [^18^F]FET BTV, which was 1.87 ± 1.10 (range: 0.30–4.30). The mean Dice similarity coefficient between the [^68^Ga]Ga-PSMA-617 BTV and the [^18^F]FET BTV was 0.58 ± 0.18 (range: 0.34–0.83), demonstrating a non-negligible mismatch between the BTV margins delineated by the two tracers. Additionally, a qualitative evaluation of the SUV images revealed different patterns of uptake of the two radiotracers, with hotspots of [^68^Ga]Ga-PSMA-617 uptake not always corresponding to hotspots of [^18^F]FET uptake or T_1_-weighted contrast-enhancing tumour.

There was a positive correlation (Pearson coefficients r: 0.50, 0.61, 0.48, 0.52, 0.32, 0.59, 0.77, 0.51, 0.30; *p* < 0.0001) between voxel-wise [^68^Ga]Ga-PSMA-617 SUV values and [^18^F]FET SUV values within the overlapping volume between [^68^Ga]Ga-PSMA-617 BTV and [^18^F]FET BTV (Figure 2).

### 2.3. Comparison between [^68^Ga]Ga-PSMA-617 and [^18^F]FET Tumour Specificity

Results of the comparison of TBR_mean_ and TBR_max_ in the BTV between [^68^Ga]Ga-PSMA-617 and [^18^F]FET are shown in Figure 3. The mean value of TBR_mean_ was significantly higher for [^68^Ga]Ga-PSMA-617 than for [^18^F]FET (21.3 ± 8.4 vs. 2.3 ± 0.4; *p* = 0.002; Figure 3B). The mean value of TBR_max_ was significantly higher for [^68^Ga]Ga-PSMA-617 than for [^18^F]FET (90.8 ± 47.7 vs. 4.7 ± 2.2; *p* = 0.002; Figure 3B). Similarly to what can be observed in Figure 1, hotspots of [^68^Ga]Ga-PSMA-617 TBR did not entirely correspond to hotspots of [^18^F]FET TBR or T_1_-weighted contrast-enhancing (CE) tumour (Figure 3).

The comparison between the CE tumour volume and the [^68^Ga]Ga-PSMA-617 BTV volume revealed that the [^68^Ga]Ga-PSMA-617 BTV volume covers the CE tumour volume and extends to adjacent regions of non-enhancing tumour (Figure 4A), resulting in a BTV approximately four times larger than the CE tumour volume (*p* = 0.0039; Figure 4B).

### 2.4. Evaluation of [^68^Ga]Ga-PSMA-617 Potential for Theranostics Application

Examples of [^68^Ga]Ga-PSMA-617 PET images with BTV and control volume of interest (VOI) segmentations are illustrated in Figure 5A. The measurements of the [^68^Ga]Ga-PSMA-617 SUV_mean_ and SUV_max_ in the BTV and in the control VOI are collected in Table 2 and Table 4. The [^68^Ga]Ga-PSMA-617 SUV_mean_ and SUV_max_ in the BTV were 1.68 ± 0.41 and 7.04 ± 2.13, respectively. The [^68^Ga]Ga-PSMA-617 SUV_mean_ and SUV_max_ in the control VOI were 0.11 ± 0.05 and 0.80 ± 0.64, respectively. [^68^Ga]Ga-PSMA-617 PET images of the two patients who undertook full-body PET scans are illustrated in Figure 5B. Patient 1 had values of [^68^Ga]Ga-PSMA-617 TLR_max_ and TLR_mean_ of 1.07 and 0.68, and patient 2 had values of TLR_max_ and TLR_mean_ of 1.20 and 0.91. TSG_mean_ and TSG_max_ values are reported in Table 5. The cohort of 10 patients had mean values of TSG_mean_ of 0.25 ± 0.12 and mean values of TSG_max_ of 0.34 ± 0.11.

## 3. Discussion

Given the ever-growing need for more selective, more specific and more affordable imaging biomarkers for recurrent GBM diagnosis and secondary treatment planning, we sought to investigate the role of [^68^Ga]Ga-PSMA-617 in the management of recurrent GBM patients. The results of our volumetric evaluations performed to test [^68^Ga]Ga-PSMA-617 tumour selectivity demonstrated that [^68^Ga]Ga-PSMA-617 accumulates in large parts of the tumour that extend beyond the [^18^F]FET-avid margins, suggesting that [^68^Ga]Ga-PSMA-617 targets a complementary biological process to [^18^F]FET, and it might be a useful diagnostic marker to delineate parts of the recurrent tumour that are neoangiogenic, but not extremely metabolically active yet. This information could be potentially used by clinicians for early treatment response assessment and to delineate regions of progressing tumour that could be targeted with external beam radiotherapy. One limitation of our study is the potential tumour growth within the time interval between the [^18^F]FET and the [^68^Ga]Ga-PSMA-617 PET scans, which could have influenced the observation of a larger [^68^Ga]Ga-PSMA-617 BTV than [^18^F]FET BTV. However, the different patterns of [^68^Ga]Ga-PSMA-617 uptake with hotspots outside the [^18^F]FET-avid and/or CE tumour support our hypothesis that the two tracers target complementary tumour development processes. In our study design we tried to mitigate the risk of tumour growth during the interval between the PET scans by scheduling patients for the [^68^Ga]Ga-PSMA-617 PET scan in the same week as the [^18^F]FET scan. This is reflected in the observed median interval of 5 days, with only two patients receiving the second scan at 13 and 15 days after their [^18^F]FET scan. It is worth noting that further reducing the interval between PET scans remains a challenge for future similar studies, due to diagnostic radioactivity dose limits for the patients, necessary radiotracers clearance time and logistics linked to radiotracer production/shipping and PET scanner and patients’ availability.

Additionally, while previous histological studies failed to establish a clear correlation between the level of PSMA expression and tumour grade across samples from different patients [3,4,5], to our knowledge this study is the first to assess the intra-patient correlation between levels of PSMA expression and levels of metabolic activity within the tumour lesion in recurrent GBM patients. The correlation results in our study suggested that more metabolically active regions within the recurrent tumour lesion tend to express higher levels of PSMA, as they have increased neoangiogenesis, which reflects what has previously been observed in treatment naïve high-grade glioma patients [17]. This observation is consistent with the hypothesis that a more metabolically active phenotype of recurrent GBM tumour is driven by a hypoxic environment that, in turn, leads to the upregulation of neoangiogenic pathways to meet the metabolic demand of the cancer cells [18,19,20].

Furthermore, as tumour grade is inversely associated with clinical outcome, it would be interesting to evaluate the correlation between [^68^Ga]Ga-PSMA-617 uptake and clinical outcome. While the present study collected imaging data for only ten patients and did not include the collection of clinical outcome data, increasing the sample size of the study and including clinical outcome data collection could help in determining the role of [^68^Ga]Ga-PSMA-617 PET as a prognostic imaging biomarker, which could be used to support early clinical decision making on treatment options at recurrence.

The results of our study revealed that [^68^Ga]Ga-PSMA-617 has a higher tumour-to-normal tissue uptake than [^18^F]FET, suggesting that [^68^Ga]Ga-PSMA-617 has a better tumour specificity than [^18^F]FET. This result highlights [^68^Ga]Ga-PSMA-617′s advantage as a diagnostic agent for recurrent tumour volume delineation for secondary treatment interventions. This observation is in agreement with the observations reported in recent studies in recurrent GBM patients [8,9,12].

It is worth noting that this finding could have significant impact in extending access to state-of-the-art diagnostics to GBM patients in regional areas of Australia lacking reliable access to either an on-site cyclotron facility and radiochemistry expertise or [^18^F]FET distribution. While cyclotron-produced fluorine-18 is almost certainly cheaper than gallium-68, [21] the radiochemical synthesis required to produce [^18^F]FET is significantly more complex and radiochemists are uncommon outside large population centres in Australia. This, coupled with regulations limiting transport of radiopharmaceuticals out of state, results in limited commercial suppliers in Australia. Limited production runs result in a delay of up to two weeks when ordering [^18^F]FET, while [^18^Ga]Ga-PSMA can be produced daily if required. Gallium-68 generators have multiple applications, largely in prostate cancer staging, which attracts a Medicare rebate in Australia. This improves the economics of using a gallium-68 generator. Transport costs of the radiopharmaceutical therefore become a significant proportion of the overall cost per scan when using [^18^F]FET. As evidence, due to the transport requirements, the distance and the urgency required, the cost of obtaining [^18^F]FET was doubled compared to the cost of generating [^18^Ga]Ga-PSMA in our study.

When comparing the results of our study with the previous literature, we observed that the values of [^68^Ga]Ga-PSMA-617 TBR_max_ in our study are approximately six times larger than the values previously determined with the [^68^Ga]Ga-PSMA-11 tracer by Sasikumar et al. [8,9] (median TBR_max_ = 14.4, range: 4.07–29.4; N = 9), but within the range of values recently determined by Kunikowska et al. [12] (median TBR_max_ = 96.7, range: 32.2–357.5; N = 15). The higher values obtained in the present study compared to those reported in the study published by Sasikumar et al. [8,9] might be explained both by the different method for background activity region selection and by the superior binding affinity of [^68^Ga]Ga-PSMA-617 compared to [^68^Ga]Ga-PSMA-11, as previously reported [10]. The lack of reported TBR_mean_ values in previously published studies limits a more thorough comparison, given that this parameter is considered more indicative than TBR_max_ [22]. The values of [^18^F]FET TBR_mean_ and TBR_max_ obtained in our study are within the range of values previously determined by Lapa et al. [23] (mean TBR_mean_ = 3.7 ± 1.8, range: 2.0–10.8; mean TBR_max_ = 3.2 ± 1.6, range: 1.8–9.5; *N* = 22) and by Pöpperl et al. [24] (mean TBR_max_ = 3.55 ± 1.07; *N* = 30).

The accumulation of [^68^Ga]Ga-PSMA-617 in regions that extend beyond the CE tumour observed in our study suggests that [^68^Ga]Ga-PSMA-617 might accumulate in regions of early neoangiogenesis that are yet to progress to a stage where they present blood–brain barrier leakage. While this observation is limited to only nine patients in our study, validating this hypothesis in a larger cohort of patients through histological analysis would support our previous observation that [^68^Ga]Ga-PSMA-617 might be a better and more comprehensive imaging biomarker for delineation of progressing tumour volume. Validation of this hypothesis with resection specimens was not possible in our study due to the poor performance status of the patients.

The values of [^68^Ga]Ga-PSMA-617 SUV_mean_ and SUV_max_ observed in the BTV and in the control VOI of this small cohort of patients are within the range of values previously reported by Kunikowska et al. [12] and by Sasikumar et al. [8] for [^68^Ga]Ga-PSMA-11 uptake in recurrent GBM patients. The [^68^Ga]Ga-PSMA-617 SUV_mean_ and SUV_max_ measurements in the BTV, the values of TLR_max_ < 1.5 obtained for the two TLR-evaluable patients in our study and the values of TSG_max_ < 1.0 suggest that PSMA-617 might not a suitable ligand for theranostics applications with β-emitters, such as lutetium-177. Since evaluation of the theranostic application of [^68^Ga]Ga-PSMA-617 has a retrospective, exploratory aim, we recognise the limitation that our observation is based on TLR measurements obtained from only two patients. As such, our results are insufficient to ultimately exclude PSMA-617 for theranostics applications with β-emitters. Nonetheless, theranostics applications of PSMA-617 radiolabelled with α-emitters might be more promising and are not to be prematurely excluded, as threshold values of SUV_mean_ in the BTV required when using α-emitters are still unknown.

An important aspect to consider when comparing results from the present study performed with [^68^Ga]Ga-PSMA-617 and previous studies using [^68^Ga]Ga-PSMA-11 is the different biodistribution and pharmacokinetics of the two tracers. In a study evaluating the biodistribution and pharmacokinetics of [^68^Ga]Ga-PSMA-617 in prostate cancer patients, Afshar-Oromieh et al. [10] demonstrated that [^68^Ga]Ga-PSMA-617 has lower accumulation into normal organs (including the liver) and slower pharmacokinetics than [^68^Ga]Ga-PSMA-11, resulting in better lesion contrast and tracer accumulation at 3 h post-injection, rather than at 1 h post-injection. A limitation of our study is that [^68^Ga]Ga-PSMA-617 PET images were acquired approximately 45 min after tracer injection. Therefore, to obtain more representative SUV and TLR values in future, larger size studies evaluating the theranostics potential of [^68^Ga]Ga-PSMA-617, PET images should be acquired 3 h post tracer injection.

## 4. Materials and Methods

### 4.1. Clinical Trial Information

This study was a diagnostic, non-randomised, uncontrolled, open-label, single-centre, single-arm, bio-availability, phase I/II pilot clinical trial, enrolling recurrent GBM patients. The study was approved by the institutional review board, and all subjects signed an informed consent. Ethics approval was obtained from the Bellberry Human Research Ethics Committee (2017-11-885). The trial was registered to the Australian New Zealand Clinical Trial Registry as ACTRN12618001346268. The study began in October 2018 and was completed in September 2021.

### 4.2. Eligibility Criteria

Male and female patients aged 18 years or older experiencing recurrence of a previously histologically confirmed glioblastoma (WHO grade IV) at resection were recruited. Qualifying patients had tumour progression noted on a pre-enrolment MRI scan, had an Eastern Cooperative Oncology Group performance status score of 2 or less, were available for scanning on two separate days and provided written informed consent for participation in this trial once eligibility was met. Women who were pregnant or lactating and patients geographically remote from the treating centre, which would inhibit multiple presentations for imaging, were excluded from this study.

### 4.3. Study Plan

Participants were required to have two PET imaging scans of approximately 90 min each on separate days within a two-week period. PET imaging was performed by qualified Nuclear Medicine Specialists. On the day of the first scan, a dose of 250 MBq of [^18^F]FET was injected intravenously approximately 20 min prior to the PET scan acquisition. [^18^F]FET PET was used as reference standard to define the extent of gross tumour mass and levels of tumour metabolic activity, as previously established [24,25,26,27]. Upon confirmation of positive [^18^F]FET uptake in the tumour via PET scan, the patient was enrolled for a [^68^Ga]Ga-PSMA-617 PET scan on a separate day. [^68^Ga]Ga-PSMA-617 PET imaging involved an intravenous injection of a dose of 150 MBq of [^68^Ga]Ga-PSMA-617 approximately 45 min prior to PET scan acquisition. Pharmacodynamic studies were not performed. T1CE MR images acquired at other sites at the time of diagnosis of tumour progression prior to enrolment in the trial were not included in the data collection plan of the trial, but they were retrospectively collated if provided by the original site of diagnosis.

### 4.4. Radiolabeling and Positro Emission Tomography Imaging

Doses of radiolabeled [^18^F]FET were purchased from Cyclotek (Sydney, Australia). Gallium-68 radiolabeling was performed on-site using an Eckert and Ziegler (Atlanta, GA, USA) gallium-68 synthesis unit. Readers should refer to the validated process outlined in the Eckert and Ziegler (Atlanta, GA, USA) “User Manual for Synthesis of gallium-68 conjugated peptides with Modular Lab eazy” for the radiosynthesis method, synthesis of ^68^Ga PSMA HBED and synthesis preparation with ABX reagents kit EZ 102. Radiochemical yield and purity of both tracers were determined by thin layer chromatography and high-performance liquid chromatography. Doses were administered if the radiochemical purity was >95%. All nuclear medicine scans were performed using a PET-CT scanner (Siemens Biograph mCT, Siemens Healthcare Pty. Ltd., Bayswater, Australia). For [^18^F]FET, a brain PET-CT scan was performed 20 min post intravenous injection of the tracer. For [^68^Ga]Ga-PSMA-617, the patients first received a brain scan at 45 min post intravenous injection of the tracer, followed by a separate vertex to mid-thigh scan at 1 h post injection, as per standard protocol at the clinical site. For all scans, a 10 min PET image was acquired followed by a computed tomography (CT) image (acquisition parameters: 3 mm slice (0.6 mm × 0.6 mm), pitch 0.8, kV 120) for attenuation correction and co-registration to the T1CE images. The PET and CT images were reconstructed with PET syngo VE60A (Siemens Healthcare Pty. Ltd., Bayswater, Australia), correcting for attenuation and gallium-68 and fluorine-18 detector efficiency. Corrected PET reconstruction parameters were: TrueX + TOF (ultraHD-PET), iterations 2, subsets 21, Gaussian filter, FWHM 2 mm; uncorrected PET reconstruction parameters were: Iterative TOF, iterations 3, subsets 21, Gaussian filter, FWHM 2 mm; ACCT Brain 3.0 I30f3 reconstruction algorithm parameters were: I30f medium smooth, 3 mm slice, increment 2 mm, window cerebrum. CT Brain 2.0I31f3 reconstruction algorithm parameters were: I31f medium smooth, 2 mm slice, increment 1 mm, window cerebrum.

### 4.5. Image Analysis

All imaging and clinical patient data were pseudo-anonymized prior to data transfer to the image analyst. Image analysis was performed by an experienced neuroimaging analyst (with 6+ years of experience in neuroimaging analysis) and was revised by a radiation oncologist (with 15+ years of experience in PET imaging). The image analysis pipeline involved image preprocessing, regions of interest segmentation and metrics calculation. Each step is detailed below. Image analysis was performed in Python using tools from the image analysis package SimpleITK [28,29,30]. The code developed for image analysis is available on our public GitHub repository at the following link https://github.com/cbri92/Genesis-GBM-001 (accessed on 10 November 2023) [31].

#### 4.5.1. Pre-Processing

DICOM images were converted into NIFTI format using dcm2niix [32]. The radioactivity concentrations in the PET images were decay corrected to the point of tracer injection, using a gallium-68 half-life of 67.71 min and a fluorine-18 half-life of 109.77 min, according to the following method. Values of activity were converted into SUV according to Equation (1) [33]:(1)SUV=Activityconcentration×BodyweightInjecteddose×eln2t12(Timeinterval),
where SUV are in units of g mL^−1^, the Activity concentration represents the values of activity in the PET image field of view and is in units of Bq mL^−1^, Body weight is in units of g, Injection dose is in units of Bq, t_1/2_ is the radiotracer half-life and is in units of min and Time interval is the time passed between the injection of the radiotracer and the PET scan acquisition and is in units of min. Assuming an average body mass density of 1 g/mL, SUV becomes a unitless quantity. The conversion of the units of activity intensity in the [^18^F]FET PET images was only completed for 9 patients, as for one patient information on injected dose, injection time, and scan time were not provided. T1CE, [^18^F]FET PET, [^68^Ga]Ga-PSMA-617 PET and [^68^Ga]Ga-PSMA-617 CT images were then registered using 3D Euler rigid transformations (six degrees of freedom) to the [^18^F]FET CT image (optimizer: gradient descent, similarity metric: Mattes mutual information) followed by linear resampling (interpolator: trilinear) to the [^18^F]FET CT image resolution. Binary masks of the brain were obtained by performing brain extraction with the BET2 tool from FSL (Oxford, UK) on the co-registered T1CE images [34]. Brain masks were then used to extract the [^18^F]FET and [^68^Ga]Ga-PSMA-617 PET images.

#### 4.5.2. Segmentation

Binary masks of the BTV and of the normal brain tissue in the contralateral part of the brain (control VOI) were delineated on the brain-extracted [^18^F]FET and [^68^Ga]Ga-PSMA-617 PET images using a previously developed semiautomated method, selecting an initial SUV threshold of 2.2 and 1.5 and a TBR threshold of 1.7 and 4 for [^18^F]FET and [^68^Ga]Ga-PSMA-617, respectively [35]. The SUV threshold value for [^18^F]FET was chosen as it had been previously determined as a cut-off threshold for identification of recurrent glioma [24,25]. Given the lack of reference values in the literature, the initial threshold value for [^68^Ga]Ga-PSMA-617 SUV was chosen arbitrarily. The TBR threshold value for [^68^Ga]Ga-PSMA-617 was chosen based on the cutoff threshold reported for recurrent gliomas in a previous study [9]. A binary mask of the overlapping [^18^F]FET BTV and [^68^Ga]Ga-PSMA-617 BTV was also delineated. A binary mask of the salivary glands was contoured on the [^68^Ga]Ga-PSMA-617 PET image. For patients with full-body [^68^Ga]Ga-PSMA-617 PET scans available, a binary mask of the liver was contoured on the full-body [^68^Ga]Ga-PSMA-617 PET image. CE tumour volume was delineated on the T1CE MRI image. All contours were reviewed by a radiation oncologist.

#### 4.5.3. Metrics Calculation

The SimpleITK package [28] in Python was used to calculate BTV volume, mean value of SUV (SUV_mean_) within the BTV and control VOI and maximum value of SUV (SUV_max_) within the BTV from the [^18^F]FET and [^68^Ga]Ga-PSMA-617 PET images. PET images were divided by the SUV_mean_ of the respective control VOI to generate TBR images. Mean and maximum TBR values (TBR_mean_ and TBR_max_) were calculated within the BTV from the [^18^F]FET and [^68^Ga]Ga-PSMA-617 TBR images. [^68^Ga]Ga-PSMA-617 TLR_mean_ and TLR_max_ were calculated by dividing the value of SUV_mean_ in the BTV by the value of SUV_mean_ in the liver, and the value of SUV_max_ in the BTV by the value of SUV_max_ in the liver, respectively. [^68^Ga]Ga-PSMA-617 TSG_mean_ and TSG_max_ were calculated by dividing the value of SUV_mean_ in the BTV by the value of SUV_mean_ in the salivary glands, and the value of SUV_max_ in the BTV by the value of SUV_max_ in the salivary glands, respectively. Tumour selectivity of [^68^Ga]Ga-PSMA-617 was assessed by comparing the tumour volume and measuring the volumetric ratio and the Dice similarity coefficient between the [^68^Ga]Ga-PSMA-617 BTV and the [^18^F]FET BTV, and by evaluating the voxel-wise correlation between [^68^Ga]Ga-PSMA-617 SUV values and [^18^F]FET SUV values within the overlapping volume between [^68^Ga]Ga-PSMA-617 BTV and [^18^F]FET BTV. Tumour specificity of [^68^Ga]Ga-PSMA-617 was assessed by comparing values of TBR_mean_ and TBR_max_ between [^68^Ga]Ga-PSMA-617 and [^18^F]FET, and by comparing the CE tumour volume delineated on T1CE image and the [^68^Ga]Ga-PSMA-617 BTV volume, when T1CE images were available. The theranostics potential of [^68^Ga]Ga-PSMA-617 was evaluated by measuring values of [^68^Ga]Ga-PSMA-617 SUV_mean_ and SUV_max_ in the BTV and in the control VOI, and by measuring [^68^Ga]Ga-PSMA-617 TLR_mean_, TLR_max_, TSG_mean_ and TSG_max_.

### 4.6. Statistical Analysis

Statistical analysis was carried out using GraphPad Prism 7 Software (Boston, MA, USA). Descriptive statistics, including group median, range, mean and standard deviation, were calculated for the volumetric measurements and values of SUV_mean_, SUV_max_, TBR_mean_ and TBR_max_. Wilcoxon matched-pairs signed-rank test, α = 0.05, was used to determine significance in the comparison of the values of TBR_mean_ and TBR_max_ between the [^68^Ga]Ga-PSMA-617 and [^18^F]FET, and in the comparison between CE tumour volume and [^68^Ga]Ga-PSMA-617 BTV. Two-tailed Pearson correlation coefficient r, α = 0.05, was evaluated in Python and was used to assess the linear correlation between voxel-wise [^18^F]FET and [^68^Ga]Ga-PSMA-617 values of SUV in the overlapping BTV. For consistency with previous studies [12,36] [^68^Ga]Ga-PSMA-617 TLR_max_ = 1.5 and TSG_max_ = 1.0 were considered the minimum cutoff threshold required to qualify PSMA-617 for application in radionuclide therapy for recurrent GBM patients.

## 5. Conclusions

Overall, this study demonstrated that [^68^Ga]Ga-PSMA-617 could find application as a diagnostic agent for recurrent tumour delineation and secondary treatment planning. Our findings could significantly expand access to state-of-the-art GBM diagnostics, as [^68^Ga]Ga-PSMA-617 could be more widely available than [^18^F]FET PET, especially in rural and regional areas of Australia that lack reliable access to [^18^F]FET distribution.

## Figures and Tables

**Figure 1 ijms-24-16208-f001:**
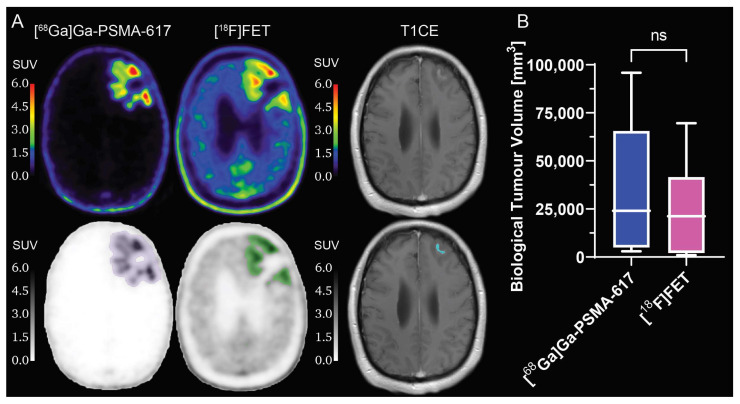
Volumetric comparison of biological tumour volume delineated on [^68^Ga]Ga-PSMA-617 and [^18^F]FET PET images. (**A**) Example of [^68^Ga]Ga-PSMA-617, [^18^F]FET PET and T1CE images of a patient (top), with biological tumour volume segmentations (bottom: purple for [^68^Ga]Ga-PSMA-617, green for [^18^F]FET and blue for T1CE); (**B**) box-and-whiskers plot of the comparison of [^68^Ga]Ga-PSMA-617 and [^18^F]FET biological tumour volume, with bar representing the median of the population. ns = no significant difference. PET: positron emission tomography. T1CE: T_1_-weighted contrast-enhanced. *N* = 10.

**Figure 2 ijms-24-16208-f002:**
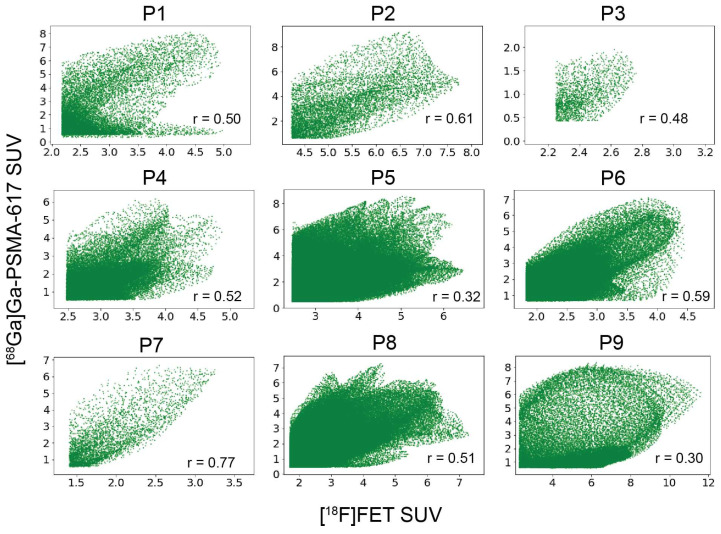
Intra-patient correlation between [^68^Ga]Ga-PSMA-617 and [^18^F]FET standardised uptake value. Voxel-wise correlation between [^68^Ga]Ga-PSMA-617 and [^18^F]FET SUV within the portion of [^68^Ga]Ga-PSMA-617 biological tumour volume overlapping the [^18^F]FET biological tumour volume. r represents the Pearson’s correlation coefficient. SUV: standardised uptake value.

**Figure 3 ijms-24-16208-f003:**
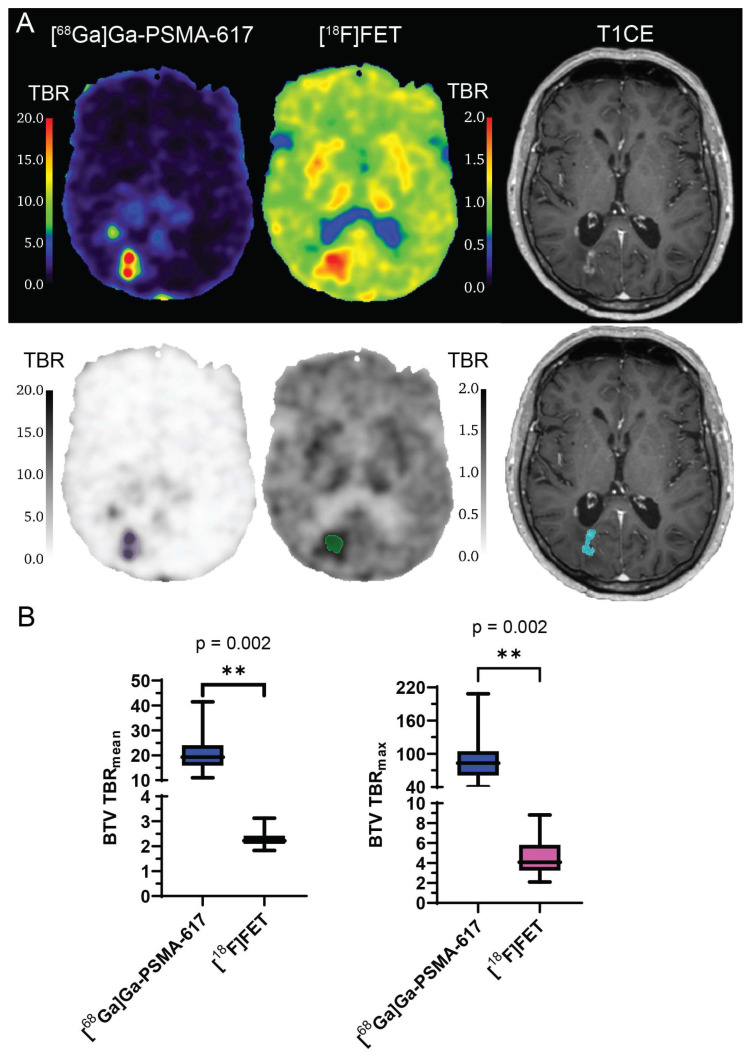
Comparison between [^68^Ga]Ga-PSMA-617 and [^18^F]FET tumour-to-brain ratio in biological tumour volume. *(***A**) Example of [^68^Ga]Ga-PSMA-617 TBR, [^18^F]FET TBR and T1CE images of a patient (top), with BTV segmentations (bottom: purple for [^68^Ga]Ga-PSMA-617, green for [^18^F]FET and blue for T1CE); (**B**) box-and-whiskers plots of the comparison of TBR_mean_ and TBR_max_ between the [^68^Ga]Ga-PSMA-617 BTV and the [^18^F]FET BTV, with bar representing the median of the population. ** = *p* < 0.01. BTV: biological tumour volume. T1CE: T_1_-weighted contrast-enhanced. TBR: tumour-to-brain ratio. *N* = 10.

**Figure 4 ijms-24-16208-f004:**
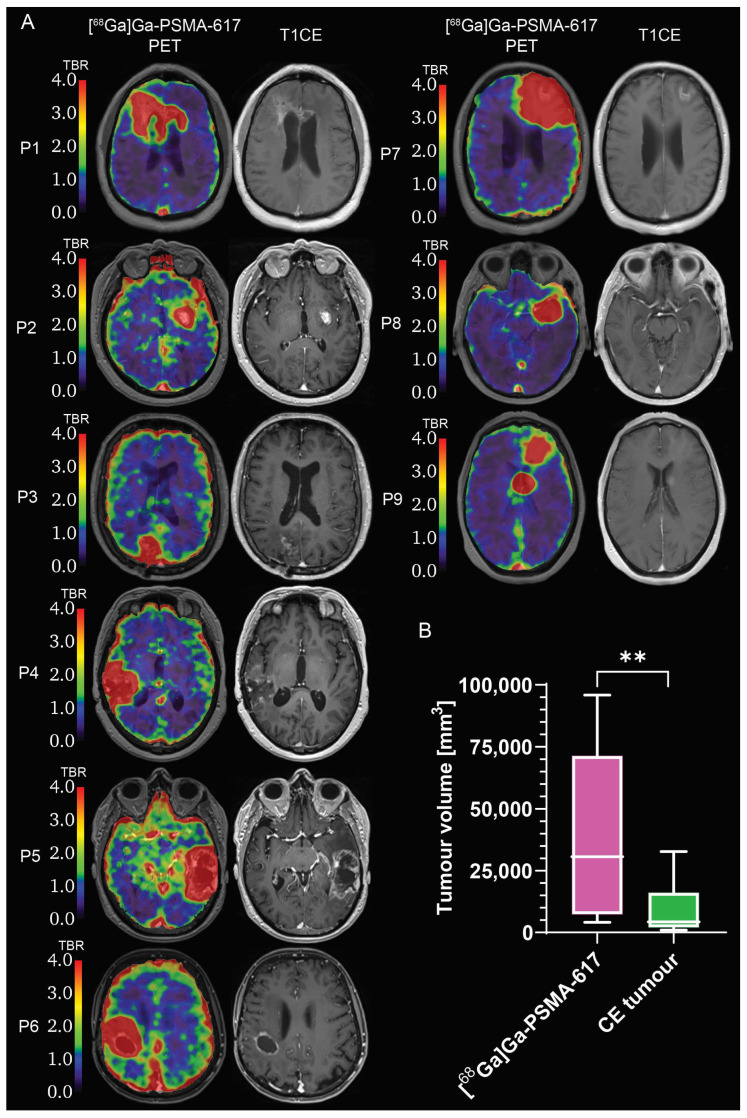
Comparison between [^68^Ga]Ga-PSMA-617 biological tumour volume and contrast-enhancing tumour volume. (**A**) [^68^Ga]Ga-PSMA-617 TBR and T1CE images; (**B**) box-and-whiskers plot of the volumetric comparison between [^68^Ga]Ga-PSMA-617 biological tumour and CE tumour, with bar representing the median of the population. ** = *p* < 0.01. CE: contrast-enhancing; PET: positron emission tomography. T1CE: T_1_-weighted contrast-enhanced. TBR: tumour-to-brain ratio. ns = no significant difference. *N* = 9.

**Figure 5 ijms-24-16208-f005:**
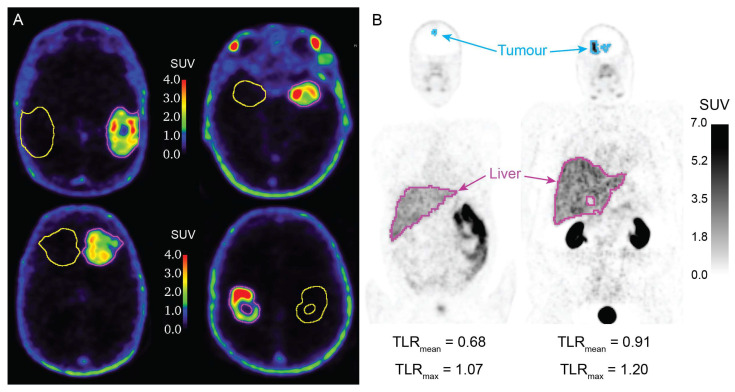
[^68^Ga]Ga-PSMA-617 PET images with segmentations. (**A**) [^68^Ga]Ga-PSMA-617 PET images of four patients, with biological tumour volume (purple) and control VOI (yellow) segmentations; (**B**) full body [^68^Ga]Ga-PSMA-617 PET images of two patients, with tumour (blue) and liver (purple) segmentations, and respective TLR_mean_ and TLR_max_ values. PET: positron emission tomography. SUV: standardised uptake value; TLR_mean_: mean tumour-to-liver ratio; TLR_max_: maximum tumour-to-liver ratio; VOI: volume of interest. *N* = 9.

**Table 1 ijms-24-16208-t001:** Dates of [^18^F]FET and [^68^Ga]Ga-PSMA-617 positron emission tomography scans.

Patient Number	[^18^F]FET PET Scan [day month year]	[^68^Ga]Ga-PSMA-617 PET Scan [day month year]	Interval between Scans [days]
1	11 October 2018	26 October 2018	15
2	9 September 2021	13 September 2021	4
3	15 November 2018	21 November 2018	6
4	29 November 2018	3 December 2018	4
5	8 November 2018	21 November 2018	13
6	13 November 2020	18 November 2020	5
7	13 November 2020	16 November 2020	3
8	24 January 2019	29 January 2019	5
9	28 March 2019	2 April 2019	5
10	15 July 2021	19 July 2021	4

**Table 2 ijms-24-16208-t002:** Results of [^68^Ga]Ga-PSMA-617 positron emission tomography metrics in the biological tumour volume.

Patient Number	Tumour Volume [mm^3^]	SUV_mean_	SUV_max_	TBR_mean_	TBR_max_
1	59,725	1.61	8.09	17.70	81.54
2	17,293	2.23	9.18	41.48	208.37
3	5070	0.79	1.96	27.39	112.88
4	4141	1.36	6.17	16.33	40.58
5	37,061	1.90	8.55	14.89	67.76
6	95,916	1.87	7.09	22.52	101.40
7	30,728	1.81	6.67	10.98	41.59
8	2956	1.68	7.28	19.23	70.85
9	83,177	1.90	8.35	22.76	98.65
10	9483	1.61	8.09	19.27	84.81

**Table 3 ijms-24-16208-t003:** Results of [^18^F]FET PET metrics in the biological tumour volume.

Patient Number	Tumour Volume [mm^3^]	SUV_mean_	SUV_max_	TBR_mean_	TBR_max_
1	39,765	-	-	2.26	4.86
2	9324	2.71	4.96	2.12	3.89
3	2080	5.36	7.73	2.14	3.08
4	974	2.41	2.76	1.83	2.09
5	14,211	2.98	4.87	2.05	3.34
6	47,071	3.20	6.44	2.22	4.46
7	33,819	2.35	4.44	2.22	4.19
8	1863	1.88	3.25	2.29	3.96
9	69,664	2.58	7.27	3.12	8.81
10	28,136	3.74	11.57	2.76	8.56

**Table 4 ijms-24-16208-t004:** Results of [^68^Ga]Ga-PSMA-617 and [^18^F]FET PET metrics in the contralateral brain volume of interest.

Patient Number	[^68^Ga]Ga-PSMA-617 SUV_mean_	[^68^Ga]Ga-PSMA-617 SUV_max_	[^18^F]FET SUV_mean_	[^18^F]FET SUV_max_
1	0.04	0.13	-	-
2	0.08	0.30	1.28	1.80
3	0.05	0.14	2.51	3.38
4	0.17	0.97	1.32	1.70
5	0.09	0.71	1.50	2.18
6	0.20	2.18	1.45	2.68
7	0.09	1.00	1.12	2.42
8	0.15	1.16	0.82	1.27
9	0.10	0.62	0.96	1.97
10	0.04	0.13	1.35	2.30

**Table 5 ijms-24-16208-t005:** Measurements of [^68^Ga]Ga-PSMA-617 tumour-to-salivary glands ratio.

Patient Number	TSG_mean_	TSG_max_
1	0.32	0.49
2	0.24	0.38
3	0.40	0.36
4	0.09	0.10
5	0.32	0.42
6	0.45	0.46
7	0.15	0.27
8	0.25	0.34
9	0.15	0.31
10	0.16	0.25

## Data Availability

The data that support the findings of this study are available from GenesisCare, but restrictions apply to protect the privacy of the patients. However, pseudo-anonymized data are available from the authors upon reasonable request and with written permission of GenesisCare.

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
