# Peer review of "Comparison between [68Ga]Ga-PSMA-617 and [18F]FET PET as Imaging Biomarkers in Adult Recurrent Glioblastoma"

_ijms, 2023, doi:10.3390/ijms242216208_

Round 1
Reviewer 1 Report
Comments and Suggestions for Authors
This study explores the potential of [68Ga]Ga-PSMA-617 as a candidate ligand for targeted radionuclide therapy with α/β-emitters in patients with recurrent GBM. The article is scientifically sound and has a logical flow.
There are several questions to address:
Line 39 Ref 7. states, "Our findings indicate that [68Ga]HBED-CC-PSMA PET/ MR can be used for noninvasive assessment of PSMA expression in GBM for diagnosis and monitoring of this disease." Not "for treatment stratification." as the authors claim. Please edit.
Line 42 111-In is a SPECT imaging agent, not PET. Please correct.
Line 44 Ref. 10 states low uptake in kidneys of [ 68Ga]Ga-PSMA-617 compared to [ 68Ga]Ga-PSMA-11, not all off-target organs; please correct.
Line 45 Ref. 11 states, "The Youden index of 68Ga-PSMA-617-PET was higher than that of 18F-FDG-PET,68Ga-PSMA-617-PET had potential differential diagnostic value." not the author's version "preliminary clinical results show high values of in vivo uptake into high-grade gliomas." Please rewrite the sentence.
Line 54 Ref. 11 states 100 times higher DOSE in Grays in tumor than kidneys, not radiotracer uptake, etc. Please correct.
Line 64-65. Please clarify in what types of tumor ([18F]F-FET has much higher specificity.
Line 320 "The radioactivity concentrations in the PET images were decay corrected to the point of tracer injection using.." line 325 "Time interval is the time passed between the injection of the radiotracer and the PET scan acquisition.." It looks like the authors applied decay time correction twice. The first time before SUV calculation and the second time while applying SUV calculations.
Figure 1 B) Boxplot of the comparison of [68Ga]Ga-PSMA-617 and [18F]F-FET biological tumour volume - Mean value for [68Ga]Ga-PSMA-617 is approximately 25000 mm3. Calculating the mean BTV from Table 1 gives 34555 mm3. Figure 3 Box B) [68Ga]Ga-PSMA-617 biological tumour estimated mean BTV is ~31000 mm3. Please correct the table values or images.
Figure 2. According to the authors, the segmentation for overlapping BTV was performed using threshold SUV values 2.2 and 1.5 for [ 18F]F-FET and[68Ga]Ga-PSMA-617, respectively. However, the graphs present an explicit cutoff below those values. For example, the graph for patient 7 has Pearson's correlation coefficients calculated below 1.5 SUV for [18F]F-FET and below 2.2 for [68Ga]Ga-PSMA-617. Please explain.
Line 342 Please explain why the SUV threshold for BTV of [68Ga]Ga-PSMA-617 was chosen 1.5 and why the SUV threshold for TBR of [68Ga]Ga-PSMA-617 was chosen 4 ? Ref. 33 only determines SUV thresholds for [18F]F-FET.
Line 185 "The correlation results in our study suggested that more metabolically active regions within the recurrent tumour lesion tend to express higher levels of PSMA." It is a stretched conclusion. The author's works show the correlation between the uptake of [18F]F-FET and [68Ga]Ga-PSMA-617 in recurrent GBM tumour. Please provide more evidence for the claim or rewrite the sentence.
Line 394 Why do authors think that [68Ga]Ga-PSMA-617 could be more widely available than [18F]F-FET PET, especially in rural areas that lack access to [18F]F-FET PET distribution lines"? The Ga-68 generators have a substantial price compared to the cyclotron-produced F-18.
One of the main concerns about the study is the volume double time for GBM. It was estimated that GBM growth is within a range of 1.3%-13.5% of total volume per day. The two-week period could potentially affect the result. Please provide the data on the time difference between two PET scans per patient. And add to the discussion session how or will it affect the final results.
Thank you.
Author Response
|
Comment 1: This study explores the potential of [68Ga]Ga-PSMA-617 as a candidate ligand for targeted radionuclide therapy with α/β-emitters in patients with recurrent GBM. The article is scientifically sound and has a logical flow. There are several questions to address:
Line 39 Ref 7. states, "Our findings indicate that [68Ga]HBED-CC-PSMA PET/ MR can be used for noninvasive assessment of PSMA expression in GBM for diagnosis and monitoring of this disease." Not "for treatment stratification." as the authors claim. Please edit.
|
|
Response 1: Thank you for pointing this out. We agree with this comment. Therefore, we have modified the sentence at line 39, removing “for treatment stratification” and adding “as an imaging biomarker for GBM patients’ management”.
|
|
Comment 2: Line 42 111-In is a SPECT imaging agent, not PET. Please correct.
|
|
Response 2: We agree. Therefore, at line 42 we clarified which, amongst the radioisotopes that we listed, are used for PET imaging and which ones are used for radionuclide therapy by adding them in brackets after mentioning each application. The sentence now reads as:
“…, including gallium-68, lutetium-177, indium-111 and yttrium-90, and thus, it can be used for both PET imaging (gallium-68, yttrium-90) and radionuclide targeted therapy (lutetium-177, indium-111, yttrium-90) [10]”.
|
|
Comment 3: Line 44 Ref. 10 states low uptake in kidneys of [ 68Ga]Ga-PSMA-617 compared to [ 68Ga]Ga-PSMA-11, not all off-target organs; please correct.
|
|
Response 3: We agree. Following the reviewer’s suggestion, we modified the sentence at line 44, substituting “to off target organs” with “to the kidneys”.
|
|
Comment 4: Line 45 Ref. 11 states, "The Youden index of 68Ga-PSMA-617-PET was higher than that of 18F-FDG-PET,68Ga-PSMA-617-PET had potential differential diagnostic value." not the author's version "preliminary clinical results show high values of in vivo uptake into high-grade gliomas." Please rewrite the sentence.
|
|
Response 4: Reference 11 cited at line 45 of our manuscript (Wang, S.; Wang, J.; Liu, D.; Yang, D. The Value of 68Ga-PSMA-617 PET/CT in Differential Diagnosis between Low-Grade and High-Grade Gliomas. J. Nucl. Med. 2018, 59, 146) explicitly reports the following conclusions: “Preliminary clinical results indicated that the value of SUV of high-grade gliomas is significantly higher than low-grade gliomas by 68Ga-PSMA-617 PET/CT. It can effectively differential the low grade gliomas from high grade gliomas.” The referenced study does not include the statement “The Youden index of 68Ga-PSMA-617-PET was higher than that of 18F-FDG-PET,68Ga-PSMA-617-PET had potential differential diagnostic value.”, as mentioned by the reviewer, and it does not involve a comparison between the Youden index of 68Ga-PSMA-617-PET and 18F-FDG-PET. Therefore, we have left the sentence in our manuscript unchanged, as this reflects the conclusions reported in the reference that we cited.
|
|
Comment 5: Line 54 Ref. 11 states 100 times higher DOSE in Grays in tumor than kidneys, not radiotracer uptake, etc. Please correct.
|
|
Response 5: We agree. We believe that the reviewer referred to reference 13 at line 54. The term “radiotracer uptake” was substituted with the term “absorbed dose”.
|
|
Comment 6: Line 64-65. Please clarify in what types of tumor ([18F]F-FET has much higher specificity.
|
|
Response 6: We agree. Following the reviewer’s suggestion, we clarified that FET has much higher tumour specificity than FDG in brain cancer by adding “in brain cancers” at the end of the sentence at line 66.
|
|
Comment 7: Line 320 "The radioactivity concentrations in the PET images were decay corrected to the point of tracer injection using.." line 325 "Time interval is the time passed between the injection of the radiotracer and the PET scan acquisition.." It looks like the authors applied decay time correction twice. The first time before SUV calculation and the second time while applying SUV calculations.
|
|
Response 7: We thank the reviewer for pointing out a potential source of ambiguity in the method used for decay time correction. Decay time correction was applied only once in our method, and to remove any ambiguity we have added the clause “, according to the following method.” At the end of the sentence starting at line 320 (line 357 of the revised manuscript), which now reads as:
“The radioactivity concentrations in the PET images were decay corrected to the point of tracer injection, using a gallium-68 half-life of 67.71 min and a fluorine-18 half-life of 109.77 min, according to the following method.”
|
|
Comment 8: Figure 1 B) Boxplot of the comparison of [68Ga]Ga-PSMA-617 and [18F]F-FET biological tumour volume - Mean value for [68Ga]Ga-PSMA-617 is approximately 25000 mm3. Calculating the mean BTV from Table 1 gives 34555 mm3. Figure 3 Box B) [68Ga]Ga-PSMA-617 biological tumour estimated mean BTV is ~31000 mm3. Please correct the table values or images.
|
|
Response 8: The plots in Figure 1B and Figure 4B (Figure 4B reports [68Ga]Ga-PSMA-617 biological tumour volume, Figure 3B reports instead TBR values in the biological tumour volume) are box and whiskers plots, and, as such, by definition, they report the minimum, the maximum, the sample median, and the first and third quartiles of the population. The bar displayed in the middle of the box represents the median, not the mean of the population.
For Figure 1B, the median value displayed for the [68Ga]Ga-PSMA-617 biological tumour volume is 24011 mm3, which corresponds to the same value obtained calculating the median of the values reported in Table 1.
For Figure 4B, the median value displayed for the [68Ga]Ga-PSMA-617 biological tumour volume is 30728 mm3, which corresponds to the median of the N=9 patients for which CE tumour volume measures were also available from the T1CE images. The reason why the median values displayed in Figure 1B and Figure 4B for [68Ga]Ga-PSMA-617 biological tumour volume do not correspond is because Figure 1B includes measurements of [68Ga]Ga-PSMA-617 biological tumour volume of N=10 patients (as stated at the end of the caption of Figure 1 at line 118), while Figure 4B includes measurements of [68Ga]Ga-PSMA-617 biological tumour volume of N=9 patients (as stated at the end of the caption of Figure 4 at line 148).
In order to clarify that the bar of the plots in Figures 1, 3 and 4 represent the median value and avoid any ambiguity, the term “median” was added to the captions of Figures 1, 3 and 4 at lines 121, 145 and 151, respectively, of the revised manuscript.
|
|
Comment 9: Figure 2. According to the authors, the segmentation for overlapping BTV was performed using threshold SUV values 2.2 and 1.5 for [ 18F]F-FET and[68Ga]Ga-PSMA-617, respectively. However, the graphs present an explicit cutoff below those values. For example, the graph for patient 7 has Pearson's correlation coefficients calculated below 1.5 SUV for [18F]F-FET and below 2.2 for [68Ga]Ga-PSMA-617. Please explain.
|
|
Response 9: As we explained in the method at lines 380-382 of the original manuscript, the segmentation of the [18F]FET and[68Ga]Ga-PSMA-617 BTVs was performed “using a previously developed semiautomated method, selecting an initial SUV threshold of 2.2 and 1.5, and a TBR threshold of 1.7 and 4 for [18F]FET and [68Ga]Ga-PSMA-617, respectively [35].” We would like to bring to the attention of the reviewer the iterative nature of the segmentation method used, which ultimately uses the TBR thresholds of 1.7 ([18F]FET) and 4 ([68Ga]Ga-PSMA-617) for defining the BTV, in order to avoid biases due to the physiological tracer uptake in the anatomical site of the tumour, as discussed into details in reference 35. Thus, it is possible that some voxels in the overlapping BTV might have a TBR value above the defined thresholds (i.e. 1.7 for [18F]FET and 4 for [68Ga]Ga-PSMA-617), but a SUV value below the initial threshold. This can occur in voxels in which the physiological uptake of the tracer in the contralateral healthy brain structure is intrinsically high.
|
|
Comment 10: Line 342 Please explain why the SUV threshold for BTV of [68Ga]Ga-PSMA-617 was chosen 1.5 and why the SUV threshold for TBR of [68Ga]Ga-PSMA-617 was chosen 4 ? Ref. 33 only determines SUV thresholds for [18F]F-FET.
|
|
Response 10: We would like to point the reviewer’s attention to the explanation of the choice of SUV and TBR thresholds chosen for [68Ga]Ga-PSMA-617, which we included in the original manuscript at lines 347-350 (lines 384-387 of the revised manuscript):
“Given the lack of reference values in the literature, the initial threshold value for [68Ga]Ga-PSMA-617 SUV was chosen arbitrarily. The TBR threshold value for [68Ga]Ga-PSMA-617 was chosen based on the cutoff threshold reported for recurrent gliomas in a previous study [9].”
|
|
Comment 11: Line 185 "The correlation results in our study suggested that more metabolically active regions within the recurrent tumour lesion tend to express higher levels of PSMA." It is a stretched conclusion. The author's works show the correlation between the uptake of [18F]F-FET and [68Ga]Ga-PSMA-617 in recurrent GBM tumour. Please provide more evidence for the claim or rewrite the sentence.
|
|
Response 11: The strong specific correlation between active tumour metabolism and PSMA expression in de novo high-grade glioma patients has been previously reported by Traub-Weidinger et al. (Traub-Weidinger, T.; Poetsch, N.; Woehrer, A.; Klebermass, E.-M.; Bachnik, T.; Preusser, M.; Mischkulnig, M.; Kiesel, B.; Widhalm, G.; Mitterhauser, M.; et al. PSMA Expression in 122 Treatment Naive Glioma Patients Related to Tumor Metabolism in 11C-Methionine PET and Survival. J. Pers. Med. 2021, 11, 624. https://doi.org/10.3390/ jpm11070624), which report the following conclusions: “significantly higher numbers of PSMA staining vessels were found in tumors with high amino acid metabolic rates. Vascular PSMA expression in gliomas was seen as a high-grade specific feature associated with elevated amino acid metabolism and short survival.”
Upon the reviewer’s suggestion, we have now included the study by Traub-Weidinger et al. (reference 17 in the revised manuscript) as supporting evidence for our claim that a similar correlation seems to exist in recurrent GBM patients. We have therefore added the clarification “in recurrent GBM patients” at the end of the sentence at line 203 of the revised manuscript, and we have modified the sentence of our revised manuscript at lines 203-206, which now reads as:
“The correlation results in our study suggested that more metabolically active regions within the recurrent tumour lesion tend to express higher levels of PSMA, as they have increased neoangiogenesis, which reflects what has previously been observed in treatment naïve high-grade glioma patients [17].”
|
|
Comment 12: Line 394 Why do authors think that [68Ga]Ga-PSMA-617 could be more widely available than [18F]F-FET PET, especially in rural areas that lack access to [18F]F-FET PET distribution lines"? The Ga-68 generators have a substantial price compared to the cyclotron-produced F-18.
|
|
Response 12: We thank the reviewer for pointing out the need to clarify our observation regarding the wider availability of [68Ga]Ga-PSMA-617 than [18F]FET PET in rural areas to improve our manuscript.
We have contextualised our observation by adding the following paragraph in the Discussion at lines 224-239 of the revised manuscript:
“It is worth noting that this finding could have significant impact in extending access to state-of-the-art diagnostics to GBM patients in regional areas of Australia lacking reliable access to either a on-site cyclotron facility and radiochemistry expertise or [18F]FET distribution. While cyclotron produced fluorine-18 is almost certainly cheaper than galli-um-68,[21] the radiochemical synthesis required to produce [18F]FET is significantly more complex and radiochemists are uncommon outside large population centres in Australia. This, coupled with regulations limiting transport of radiopharmaceuticals out of state, results in limited commercial suppliers in Australia. Limited production runs result in a delay of up to two weeks when ordering [18F]FET, where [18Ga]Ga-PSMA can be produced daily if required. Gallium-68 generators have multiple applications, largely in prostate cancer staging, which attracts a Medicare rebate in Australia. This improves the economics of using a gallium-68 generator. Transport costs of the radiopharmaceutical therefore become a significant proportion of the overall cost per scan when using [18F]FET. As evidence, due to the transport requirements, the distance and the urgency required, the cost of obtaining [18F]FET was doubled compared to the cost of generating [18Ga]Ga-PSMA in our study.”
Additionally, we clarified that our statement pertained to rural and regional areas of Australia by modifying the conclusion statement at lines 432-434 of the revised manuscript to:
“Our findings could significantly expand access to state-of-the-art GBM diagnostics, as [68Ga]Ga-PSMA-617 could be more widely available than [18F]FET PET, especially in rural and regional areas of Australia that lack reliable access to [18F]FET distribution.”
|
|
Comment 13: One of the main concerns about the study is the volume double time for GBM. It was estimated that GBM growth is within a range of 1.3%-13.5% of total volume per day. The two-week period could potentially affect the result. Please provide the data on the time difference between two PET scans per patient. And add to the discussion session how or will it affect the final results.
|
|
Response 13: We thank the reviewer for the suggestion. We have addressed this suggestion by adding in the Result section (Subsection 2.1) a table (Table 1) with the dates of the two scans and the interval between scans, and the following sentence at lines 87-89 of the revised manuscript:
“Dates of the PET scans for each patient are reported in Table 1. The median interval between [18F]FET and [68Ga]Ga-PSMA-617 PET scans was 5 days (range: 3-15).”
Additionally, we have added the following paragraph to the Discussion section at lines 185-198 of the revised manuscript:
“One limitation of our study is the potential tumour growth within the time interval between the [18F]FET and the [68Ga]Ga-PSMA-617 PET scans, which could have influenced the observation of a larger [68Ga]Ga-PSMA-617 BTV than [18F]FET BTV. However, the different patterns of [68Ga]Ga-PSMA-617 uptake with hotspots outside the [18F]FET-avid and/or CE tumour support our hypothesis that the two tracers target complementary tumour development processes. In our study design we tried to mitigate the risk of tumour growth during the interval between the PET scans by scheduling patients for the [68Ga]Ga-PSMA-617 PET scan in the same week as the [18F]FET scan. This is reflected in the observed median interval of 5 days, with only two patients receiving the second scan at 13 and 15 days after their [18F]FET scan. It is worth noting that further reducing the interval between PET scans remains a challenge for future similar studies, due to diagnostic radioactivity dose limits to the patients, necessary radiotracers clearance time and logistics linked to radiotracer production/shipping, PET scanner and patients’ availability.”
|
Reviewer 2 Report
Comments and Suggestions for Authors
The manuscript "Comparison between [68Ga]Ga-PSMA-617 and [18F]F-FET PET as imaging biomarkers in adult recurrent glioblastoma" by Brighi et al present and interesting comparison between PSMA- and FET-PET for the diagnosis of GBM.
The evaluation of the PET-investigations are well described and the methods used relevant for the primary objective of the study. The authors suggest some improvements for follow up studies such as a later scan time for [68Ga]Ga-PSMA-617, up to 3h post injection. Although the number of patients are few the results indicate that [68Ga]Ga-PSMA-617 might be preferred over [18F]FET for GBM diagnosis. Even less statistically significant is the finding that the TLR is too low for the use of [177Lu]Lu-PSMA-617, therapy, but still an interesting obseration. This is study should be repeated with a larger number of subjects to confirm the findings, which are clinically highly interesting.
There are only minor changes required such as correct nomenclature for [18F]F-FET and [18F]F-FDG, which should be [18F]FET and [18F]FDG (it´s not needed to add F in [18F]F-FET, since fluorine it´s part of the molecule, in contrast to [68Ga]Ga-PSMA-617 where gallium is not part of the molecule, unless labelled). In text (see line 289) gallium-68 should be used, not 68Ga or Ga-68.
Author Response
|
Comment 1: The manuscript "Comparison between [68Ga]Ga-PSMA-617 and [18F]F-FET PET as imaging biomarkers in adult recurrent glioblastoma" by Brighi et al present and interesting comparison between PSMA- and FET-PET for the diagnosis of GBM.
The evaluation of the PET-investigations are well described and the methods used relevant for the primary objective of the study. The authors suggest some improvements for follow up studies such as a later scan time for [68Ga]Ga-PSMA-617, up to 3h post injection. Although the number of patients are few the results indicate that [68Ga]Ga-PSMA-617 might be preferred over [18F]FET for GBM diagnosis. Even less statistically significant is the finding that the TLR is too low for the use of [177Lu]Lu-PSMA-617, therapy, but still an interesting observation. This is study should be repeated with a larger number of subjects to confirm the findings, which are clinically highly interesting.
There are only minor changes required such as correct nomenclature for [18F]F-FET and [18F]F-FDG, which should be [18F]FET and [18F]FDG (it´s not needed to add F in [18F]F-FET, since fluorine it´s part of the molecule, in contrast to [68Ga]Ga-PSMA-617 where gallium is not part of the molecule, unless labelled). In text (see line 289) gallium-68 should be used, not 68Ga or Ga-68.
|
|
Response 1: We thank the reviewer for the positive feedback on our manuscript. Following the reviewer’s suggestions we have corrected the nomenclature for [18F]F-FET and [18F]F-FDG to [18F]FET and [18F]FDG, respectively, throughout the manuscript (including in Tables, Figures and figure captions). Additionally, we have also corrected the nomenclature for radioisotopes, such as gallium-68, yttrium-90, lutetium-177 and indium-111, from 68Ga to gallium-68 throughout the text.
|

Round 2
Reviewer 1 Report
Comments and Suggestions for Authors
Thank the authors for the prompt and sufficed answers.